# Learning Symbolic Models for Graph-Structured Physical Mechanism

**Hongzhi Shi[1], Jingtao Ding[1], Yufan Cao[1], Quanming Yao[1], Li Liu[2], Yong Li[1]**
Department of Electronic Engineering, Tsinghua University[1]
Biren Tech[2]
`liyong07@tsinghua.edu.cn`

## Abstract

Graph-structured physical mechanisms are ubiquitous in real-world scenarios, thus revealing underneath formulas is of great importance for scientific discovery. However, classical symbolic regression methods fail on this task since they can only handle input-output pairs that are not graph-structured. In this paper, we propose a new approach that generalizes symbolic regression to graph-structured physical mechanisms. The essence of our method is to model the formula skeleton with a message-passing flow, which helps transform the discovery of the skeleton into the search for the message-passing flow. Such a transformation guarantees that we are able to search a message-passing flow, which is efficient and Pareto-optimal in terms of both accuracy and simplicity. Subsequently, the underneath formulas can be identified by interpreting component functions of the searched message-passing flow, reusing classical symbolic regression methods. We conduct extensive experiments on datasets from different physical domains, including mechanics, electricity, and thermology, and on real-world datasets of pedestrian dynamics without ground-truth formulas. The experimental results not only verify the rationale of our design but also demonstrate that the proposed method can automatically learn precise and interpretable formulas for graph-structured physical mechanisms.

## 1 Introduction

For centuries, the development of the natural sciences has been based on human intuition to abstract physical mechanisms represented by symbolic models, i.e., mathematical formulas, from experimental data recording the phenomena of nature. Among these developments, many mechanisms are naturally graph-structured (Leech, 1966), where the physical quantities are associated with individual objects (e.g., mass), pair-wise relationships (e.g., force) and the whole system (e.g., overall energy), corresponding to three types of variables on graphs: node/edge/global variables. For example, as shown in Figure 1(a), the mechanical interaction mechanism in multi-body problem corresponds to a graph with masses $(m_i)$, positions $(\vec{V_i})$ as attributes of nodes, and spring constants $(k_{ij})$ as attributes of edges, which, together with the graph connectivity, yields the acceleration as output attributes of nodes; while in the case of resistor circuit, nodes and edges correspond to voltages and resistances, respectively, and these attributes define a graph-level overall power of the circuit.

In the past few years, Symbolic Regression (SR) (Sahoo et al., 2018; Schmidt & Lipson, 2009; Udrescu et al., 2020), which searches symbolic models $y = \mathcal{F}(x)$ from experimentally obtained input-output pairs $\{(x, y)\}$ with $\mathcal{F}$ being an explicit formula, has become a promising approach trying to automate scientific discovery. Traditional SR methods include genetic programming-based methods (Schmidt & Lipson, 2009; Fortin et al., 2012) working by generating candidate formulas by "evolution" (i.e., manipulations), and deep learning-based methods (Li et al., 2019; Biggio et al., 2021; Zheng et al., 2021) utilizing sequence models to generate candidate formulas. However, these methods are designed for traditional SR problems on input-output pairs $\{(x, y)\}$ without considering graph information.

To exploit the inherent graph structure in physical mechanisms, as shown in Figure 1(b), SR on graphs aims to find a formula $\mathcal{F}$ that characterizes a mapping from input $\{\mathcal{G}, X\}$ to output $y$, with $X$ and $y$ both inside graph structure $\mathcal{G}$. To perform this, we need both fine exploitation of inherent

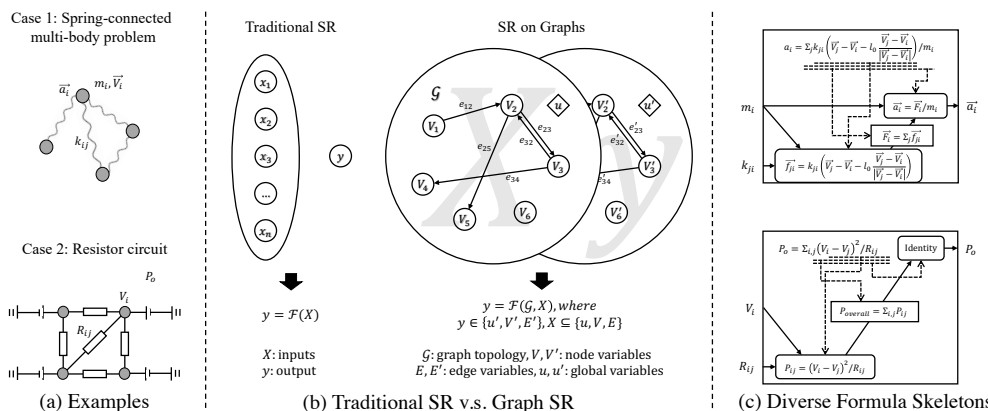

Figure 1: (a) Two examples of graph-structured physical mechanisms. (b) Illustration of traditional SR (left) and SR on graphs (right). (c) Two different formula skeletons in the two examples, where all formula components are interconnected accordingly to construct an overall formula.

graph structures of physical mechanisms and well achievement of flexibility regarding diverse forms of interaction between entities in the physical world. Graph Neural Network (GNN) has recently been incorporated into SR for discovering mechanisms behind particle interactions (Cranmer et al., 2020; Lemos et al., 2022). However, obvious setbacks exist that the message-passing flow of GNN, corresponding to the formula skeleton, required to be manually designed to learn the underlying mechanisms, is impractical because the formula skeletons usually remain unknown and are significantly different in diverse physical domains as shown in Figure 1(c).

To solve this problem, inspired by the correspondence between the skeleton and message-passing flow in GNN, our core idea is to transform the discovery of the skeleton into the search for message-passing flow, which paves the way for identifying the underneath formula by interpreting each component function in the searched message-passing flow. However, due to the coupling relationship between the skeleton and the component formula in the skeleton, neither of them can be independently identified, implying a vast, highly entangled search space for both message-passing flow and component functions. To tackle this challenge, we formulate a bi-level optimization problem that searches for the message-passing flow by pruning strategy at the upper level on condition that its component functions have been optimized with deep learning (DL) at the lower level. Besides empirical accuracy, it is equally vital but non-trivial to maintain explicit interpretability and generalization ability in discovered formulas. We propose to search the Pareto-optimal message-passing flow between accuracy and simplicity by carefully designing a scoring function involving a complexity function of message-passing flows that optimizes both aspects across different searching steps.

Our contributions can be summarized as the following three aspects,

- We generalize the problem of learning formulas with given skeletons (inductive bias) from graph data in Cranmer et al. (2020) by additionally learning the formula skeleton from data, which is essential for learning graph-structured physical mechanisms from diverse physical domains.

- We propose a novel method to learn graph-structured physical mechanisms from data without knowing the formula skeleton by searching the Pareto-optimal message-passing flows of GNN together with the symbolic models as components.

- We conduct experiments on five datasets from diverse physical domains, including mechanics, electricity, thermology, and two real-world datasets about pedestrian dynamics, demonstrating that our model can first automatically identify the correct skeleton based on collected data instead of expert knowledge and then learn the overall symbolic model for corresponding graph-structured physical mechanism.

## 2 THE PROPOSED METHOD

Before introducing the proposed method, we first formally define the the problem of symbolic regression on graphs.

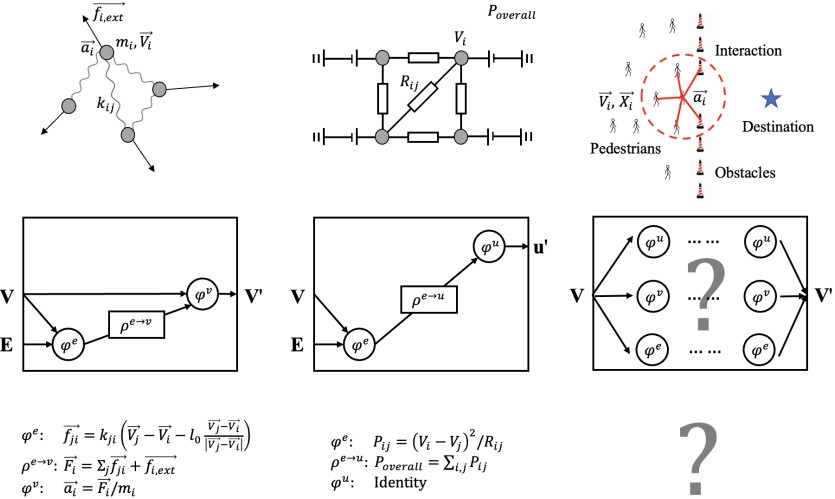

Figure 2: Examples of three different graph-structured physical mechanisms and their corresponding message-passing flows. Notice here that the three examples differ from each other greatly in the layouts of message-passing flows. Every component of the message-passing flows represents a certain function (listed at the bottom) on the graph abstracted from the physical scenarios given in the first row. (1) Left: the multi-body kinematics scenario where the skeleton is the same as in Cranmer et al. (2020); (2) Middle: the resistor circuit scenario where the skeleton is largely different from that in Cranmer et al. (2020); (3) Right: the collision avoidance scenario where the skeleton is unknown.

**Definition 1 (Variables on Graphs)** *The topology of the graph is denoted as $\mathcal{G}$, and its variables include $\{\mathbf{V}, \mathbf{E}, \mathbf{u}\}$, where $\mathbf{V}$ denotes the set of node-level variables, $\mathbf{E}$ denotes the set of edge-level variables and $\mathbf{u} \in \mathbb{R}^{n_u}$ denotes a global variable which interacts with elements in $\mathbf{V}$ and $\mathbf{E}$ through topology $\mathcal{G}$. Specifically, $\mathbf{v}_i \in \mathbb{R}^{n_v}$ is the variable associated with the i-th node while $\mathbf{e}_{ij} \in \mathbb{R}^{n_e}$ is the variable associated with the edge connecting the i-th and j-th nodes. $n_u$, $n_v$, and $n_e$ denote the dimensions of the global, node, and edge variables.*

**Definition 2 (Symbolic Regression on Graphs)** *Given a set of $\{(\mathcal{G}_i, X_i, \mathbf{y}_i)\}$, where $X_i \subset \{\mathbf{V}, \mathbf{E}, \mathbf{u}\}$, which are known variables, $\mathbf{y}_i \in \{\mathbf{V}', \mathbf{E}', \mathbf{u}'\}$, which are unknown variables, and variables with prime denote output variables, we aim to find an accurate and compact formula $\mathcal{F}(\cdot)$ that fits $\mathbf{y} = \mathcal{F}(\mathcal{G}, X)$.*

## 2.1 MODEL FORMULA SKELETON WITH MESSAGE-PASSING FLOW

As shown in Figure 1(c), a formula $\mathcal{F}$ describing graph-structured physical mechanisms can always be decomposed into several formula components, each representing its association with an individual node, pair-wise relationships between nodes or the whole system. As illustrated in the specific diagram, these components are interconnected according to the variable dependency, termed as "*skeleton*". Moreover, differences between the two examples also indicate the existence of diverse skeletons underlying different physical mechanisms.

The key insight is that the skeleton has a strong correspondence with the message-passing flows in GNN, which differs a lot in various physical scenarios, including mechanics (Sanchez-Gonzalez et al., 2020), electricity (Zhang et al., 2019), thermology (Chamberlain et al., 2021). The message-passing flows can be diverse by cascading multiple blocks, changing/removing some functions and adjusting the embedding sizes. A block of full GNN contains the updating of edges, nodes, and graph representation respectively as follows (Battaglia et al., 2018),

$$
\begin{aligned}
\mathbf{e}'_{ij} = \phi^e\left(\mathbf{e}_{ij}, \mathbf{v}_i, \mathbf{v}_j, \mathbf{u}\right) \quad &\overline{\mathbf{e}}'_i = \rho^{e \to v}\left(E'_i\right) \\
\mathbf{v}'_i = \phi^v\left(\overline{\mathbf{e}}'_i, \mathbf{v}_i, \mathbf{u}\right) \quad &\overline{\mathbf{e}}' = \rho^{e \to u}\left(E'\right) \\
\mathbf{u}' = \phi^u\left(\overline{\mathbf{e}}', \overline{\mathbf{v}}', \mathbf{u}\right) \quad &\overline{\mathbf{v}}' = \rho^{v \to u}\left(V'\right)
\end{aligned}
\tag{1}
$$

where $\phi(\cdot)$ denotes the message functions and $\rho(\cdot)$ denotes the aggregation functions. We provide three examples in different physical scenarios, as shown in Figure 2, to illustrate the well-defined analogy between message-passing flows and skeletons.

Table 1: Analogy between graph-structured formulas and GNN.

| computation steps of graph-structured formulas | architecture of message-passing networks |
| --- | --- |
| # intermediate variables | embedding size |
| dependent variables | message-passing connection |
| formulas with (multi-dimensional) values as input | message function $\phi$ |
| formulas with set as input | aggregation function $\rho$ |

**Example 1 (Mechanics: Multi-body Kinematics)** *In this problem, we aim to find the particles' acceleration. We have an edge update function $\phi^e$ since particle pairs determine string forces, while the aggregation function $\rho^{e \to v}$ of edges is based on the independent action principle of force.*

**Example 2 (Electricity: Resistor Circuit)** *The objective of this problem is to find the overall power of a given resistor circuit. An edge update function $\phi^e$ corresponds to the computation of single-resistor power utilizing Joule's Law, and an aggregation $\rho^{e \to u}$ from edge to global appears for summation to get overall power.*

**Example 3 (Pedestrian Dynamics: Collision Avoidance)** *In this problem, we aim to find the pedestrians' acceleration according to their postions and velocities. The formulas including the skeletons to describe this relationship can be diverse, which highly depends on the pedestrian scenarios.*

## 2.2 Transforming into the Task of Message-Passing Flow Searching

The message-passing flows of GNN correspond to explicit meanings in the symbolic calculation for graph-structured mechanisms, which is summarized in Table 1. This strong resemblance inspired us to devise a transformation of the primitive SR task on graphs into a relatively more practical task of searching message-passing flows. Our model has two stages: message-passing flow searching and message-passing flow-based SR. Specifically, at stage 1, we need to search the message-passing flow as the formula skeleton. Then at stage 2, we need to symbolize components into formulas and cascade them according to the skeleton to get the final graph-structured mechanisms.

With the above transformation, it is clear that we need to solve the following bi-level optimization problem in stage 1, i.e.,

$$\{\mathbf{P}^*, \mathbf{M}^*\} = \underset{\mathbf{M}, \mathbf{P}}{\arg \min}\, s(\mathbf{P}; \mathbf{M}), \tag{2}$$

where $\mathbf{M}$ denotes the message-passing flows, $\mathbf{P}$ denotes the parameters in the DL components, $\mathbf{M}^*$ and $\mathbf{P}^*$ denote the Pareto-optimal one, and $s(\cdot)$ gauges how well the finally learned formula obtained by $\mathbf{M}$ and $\mathbf{P}$ performs. However, there are two core challenges to learning formulas on graphs: (i) considering simplicity and accuracy simultaneously for graph SR is difficult; (ii) the discrete search space composed of skeletons and component formulas is prohibitively huge.

## 2.3 Searching Message-Passing Flows M (Formula Skeletons)

To deal with the first challenge, we are motivated to change equation 2 into

$$\mathbf{M}^* = \underset{\mathbf{M}}{\arg \min}\, s(\mathbf{P}^*; \mathbf{M}), \tag{3}$$

$$\text{s.t.} \quad \mathbf{P}^* = \underset{\mathbf{P}}{\arg \min}\, l(\mathbf{P}; \mathbf{M}), \tag{4}$$

where equation 3 and equation 4 are two optimization problems in the upper level and lower level, $s(\mathbf{P}^*; \mathbf{M}) = l(\mathbf{P}^*; \mathbf{M}) + \lambda c(\mathbf{M})$ is the score taking both simplicity and accuracy into consideration, $l(\cdot)$ denotes the error loss of predicting outputs (see details in Appendix B.4), $\lambda$ is the weight, and $c(\cdot)$ denotes the complexity of message-passing flow. The design of complexity $c(\cdot)$ is flexible (see details in Appendix B.5), and we calculate the complexity as follows, (i) for each layer (corresponding to a function in $\{\phi_u, \phi_v, \phi_e\}$), the complexity can be calculated as the product of the embedding size of this layer and the number of inputs in this layer; (ii) the whole complexity of the message-passing flow can be calculated as the summation of the complexity of each layer. The optimization at the lower level is solved by training parameters $\mathbf{P}$ of a DL model given the structure $\mathbf{M}$, while the

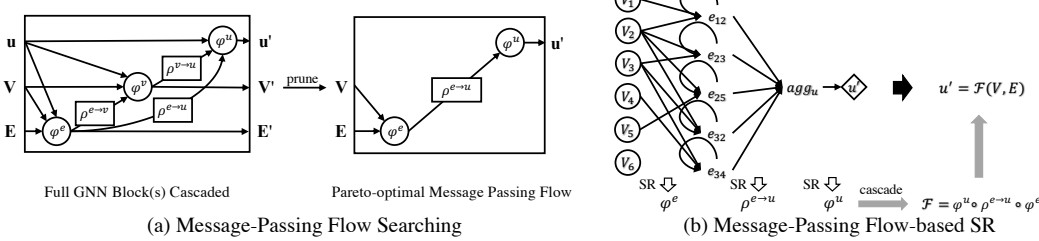

(a) Message-Passing Flow Searching

(b) Message-Passing Flow-based SR

Figure 3: The framework of our model. We partition the task into two stages: (a) Message-Passing Flow Searching and (b) Message-Passing Flow-based SR.

optimization at the upper level w.r.t. $\mathbf{M}$ is difficult because $\mathbf{M}$ forms a huge discrete search space including # blocks, # message-passing layers, connections and embedding sizes.

Another insight facilitating the dealing with the second challenge is that if the message-passing flow is a super-structure of the ground-truth one, i.e., redundant computations are done, it results in merely subtle variations of the loss. Still, if the message-passing flow is a sub-structure of the ground truth, i.e., some necessary computations are missing, the loss jumps up with a change of magnitude.

Such an observation together with equation 3 facilitates us first to search an initial message-passing flow that is the super-structure of the ground-truth and then learn to prune the message-passing flows to get both compact and expressive message-passing flows. The framework of our model is shown in Figure 3(a), which sequentially search # blocks, # layers, connections and embedding sizes in a hierarchical way and the four steps in detail are listed as follows.

**Step 1: Search Message-Passing Blocks.** First, we need to find a message-passing flow that the ground-truth message-passing flow is its sub-structure. To achieve this goal, we first stack several full message-passing blocks. By optimizing the score in equation 3, we find the Pareto-optimal number of message-passing blocks, where we take the number of blocks as the complexity in equation 3 and RMSE between the predicted value and the ground-truth as the loss.

**Step 2: Search Message-Passing Layers.** As mentioned, a full message-passing block contains three layers corresponding to updates of edge, node, and graph representation. However, not all of them are necessary for obtaining the output. To find the most compact layers, we try to delete each layer to see whether the score in equation 3 increases or decreases, where we define the complexity as the number of layers and RMSE as the loss. Our pruning-based searching method is based on the unique insight in SR, which is much more efficient than brute force search. Specifically, our method can significantly decrease the computational cost from $O(2^n)$ to $O(n)$, where $n$ is the number of initial layers.

**Step 3: Search Necessary Inputs.** We further filter out the useless inputs for each layer. Specifically, we adopt a similar strategy to the previous searching step: try to delete each input to see whether the score in equation 3 will rise or drop, where the complexity is the number of connections and the loss is RMSE. Similar to step 2, our model can significantly decrease the computational cost from $O(2^n)$ to $O(n)$, where $n$ is the number of initial inputs.

**Step 4: Search Embedding Sizes.** To ensure that the embedding in each layer is compact and with explicit physical meanings, we use the score given in equation 3 to find the Pareto-optimal embedding size for each embedding, where the complexity is defined as the embedding size and RMSE defines the loss. We try to reduce the embedding size to find the embedding size with the highest score. At the same time, we fix other embedding sizes as a large enough number to ensure the information bottleneck can only be caused by the embedding size we are searching for.

## 2.4 THE LEARNING PROCEDURE

After obtaining the message-passing flow $\mathbf{M}^*$ and the parameters of DL component function $\mathbf{P}^*$ at the first stage, we follow Cranmer et al. (2020) to symbolize each DL component into formulas, and then cascade them according to the skeleton represented in $\mathbf{M}^*$ into a whole formula at the second stage, as shown in Figure 3(b). For aggregation functions $\rho$ corresponding to set functions, (i) we choose several commonly used aggregators as candidates, including *sum*, *mean*, and *max*, while other aggregators can be generated by them, and select the maximum operator to replace the softmax function, (ii) we perform SR on input-output pairs from trained GNN component functions, (iii) we

fine-tune all constants in the overall function (given by cascading component functions), thereby avoiding the accumulation of errors.

# 3 EVALUATION

## 3.1 EXPERIMENT ON CLASSICAL PHYSICAL SCENARIOS

**Dataset.** We utilize five datasets in different physical domains to demonstrate that our model has the ability to rediscover the well-known graph-structure physical mechanisms, as introduced in Section 2.1 and Appendix A.1. We provide two cases of mechanics scenarios, one of electricity scenario and two of thermology scenarios. For both mechanics and thermology scenarios, there are two selected cases with different complexity, where the content listed in parentheses is associated with the more complex scenarios. Detailed information about formulas and data generation is reported in Appendix A.1.

**Metrics.** Given the same input, we use the coefficient of determination $R^2$, indicating the proportion of the output variation that can be predicted from the input variables. Specifically, it is calculated by the output of distilled formula and the output of the ground-truth formula to measure whether the learned formula is accurate enough. $R^2$ can be calculated as $R^2 = 1 - \sum_i (y_i - \hat{y}_i)^2 / \sum_i (y_i - \bar{y})^2$, where $\bar{y} = \sum_i y_i / n$.

**Comparing Methods.** We compare our model with learning symbolic models from deep learning with inductive bias (SymDL) (Cranmer et al., 2020) to demonstrate that our model is flexible in more scenarios and a variant of our model that uses a full graph network without pruning searching (FullGN) for the ablation study (1-layer full GNN that removes non-exist inputs and non-required outputs). The message-passing flows of SymDL and FullGN are shown in Appendix A.3.

| Method | Mechanics (simple) | | Mechanics (complex) | |
|---|---|---|---|---|
| | **Learned formulas** | **R2** | **Learned formulas** | **R2** |
| **SymDL (Cranmer, 2020)** | $a_i[0] = k_{ij}/m_i \sum_j ((C_1\Delta x_{ij} + C_2\Delta y_{ij}) + (C_3\Delta x_{ij} + C_4\Delta y_{ij})/r)$ $a_i[1] = k_{ij}/m_i \sum_j ((C_1\Delta x_{ij} + C_2\Delta y_{ij}) + (C_3\Delta x_{ij} + C_4\Delta y_{ij})/r)$ | 1.000 | $a_i[0] = \sum_j ((C_1m_i + m_j)(\Delta x_{ij} + C_2\Delta y_{ij})/r + m_i m_j(C_3\Delta x_{ij} + \Delta y_{ij})) + C_4 F_{i,ext,x}/m_i$ $a_i[1] = \sum_j ((C_1m_i + m_j)(\Delta x_{ij} + C_2\Delta y_{ij})/r * C_5 - m_i m_j(C_3\Delta x_{ij} + \Delta y_{ij})) + C_4 F_{i,ext,y}/m_i$ | 0.203 |
| **FullGN** | $a_i[0] = k_{ij}/m_i \sum_j ((C_1\Delta x_{ij} + C_2\Delta y_{ij}) + (C_3\Delta x_{ij} + C_4\Delta y_{ij})/r)$ $a_i[1] = k_{ij}/m_i \sum_j ((C_1\Delta x_{ij} + C_2\Delta y_{ij}) + (C_3\Delta x_{ij} + C_4\Delta y_{ij})/r)$ | 1.000 | $a_i[0] = \sum_j (C_1(\Delta x_{ij} + C_2\Delta y_{ij})/(m_i r) + (m_j \Delta x_{ij}/\Delta y_{ij}) + C_4 F_{i,ext,x}/m_i$ $a_i[1] = \sum_j (C_1(\Delta x_{ij} + C_2\Delta y_{ij})/(m_i r) + (\Delta x_{ij} + C_3\Delta y_{ij})/r) + C_4 F_{i,ext,y}/m_i$ | 0.529 |
| **Ours** | $a_i[0] = k_{ij}/m_i \sum_j ((C_1\Delta x_{ij} + C_2\Delta y_{ij}) + (C_3\Delta x_{ij} + C_4\Delta y_{ij})/r)$ $a_i[1] = k_{ij}/m_i \sum_j ((C_1\Delta x_{ij} + C_2\Delta y_{ij}) + (C_3\Delta x_{ij} + C_4\Delta y_{ij})/r)$ | 1.000 | $a_i[0] = k_{ij}/m_i \sum_j ((C_1\Delta x_{ij} + C_2\Delta y_{ij}) + (C_3\Delta x_{ij} + C_4\Delta y_{ij})/r) + C_5 f_{i,ext,x}/m_i + C_6 \sum_i f_{i,ext,x}/\sum_i m_i$ $a_i[1] = k_{ij}/m_i \sum_j ((C_1\Delta x_{ij} + C_2\Delta y_{ij}) + (C_3\Delta x_{ij} + C_4\Delta y_{ij})/r) + C_5 f_{i,ext,y}/m_i + C_6 \sum_i f_{i,ext,y}/\sum_i m_i$ | 0.917 |

| Method | Electricity | | Thermology (simple) | | Thermology (complex) | |
|---|---|---|---|---|---|---|
| | **Learned formulas** | **R2** | **Learned formulas** | **R2** | **Learned formulas** | **R2** |
| **FullGN** | $P_o = C_1 \sum_{i,j}(V_i - V_j)(V_i/R + C_2) + \sum_i \sum_j (V_i V_j + C_2 V_i^2 + C_3)/R$ | 0.587 | $\Delta S = C_1 \sum_{i,j} C_2(T_j \star T_i) + C_3 T_i + \sum_i \sum_j \alpha(T_i - C_4 T_i)$ | 0.318 | $\Delta S = C_1 \sum_{ij} C_3\alpha(T_j/T_i + C_4 T_i * \Delta Q_{i,ext}) + C_6 \Delta Q_{i,ext}/T_i$ | 0.189 |
| **Ours** | $P_o = C_1 \sum_{i,j}(C_2 V_i - C_3 V_j)^2/R$ | 1.000 | $\Delta S = C_1 \sum_{i,j} C_2\alpha(T_j - T_i)/T_i$ | 1.000 | $\Delta S = C_1 \sum_i \left(C_2 \sum_j C_3\alpha(T_j - T_i) + C_4 \Delta Q_{i,ext}\right)/T_i$ | 0.993 |

Figure 4: Correctness comparison based on learned formulas and $R^2$ in five scenarios. The formulas learned by our model are the same as the ground-truth formulas

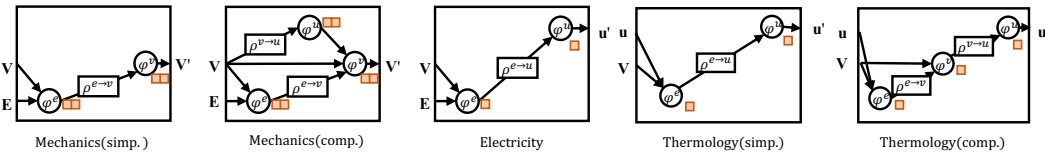

Mechanics(simp.)  Mechanics(comp.)  Electricity  Thermology(simp.)  Thermology(comp.)

Figure 5: Learned message-passing flows by our model.

**Plausibility Comparison with Baselines.** We first compare the applicability of our method with the SOTA baseline (Cranmer et al., 2020). As listed in Table 4, our method can be applied to all

Table 2: The running times (mins) for two stages in different scenarios.

| Scenario | | Mechanics (simple) | (complex) | Electricity | Thermology (simple) | (complex) |
|---|---|---|---|---|---|---|
| SymDL | | 69.91 | 70.82 | – | – | – |
| FullGN | | 86.25 | 113.59 | 59.23 | 67.21 | 69.23 |
| Ours | Stage 1 | 11.59 | 18.42 | 11.26 | 11.37 | 12.68 |
| | Stage 2 | 68.75 | 93.12 | 28.96 | 30.88 | 45.27 |
| | Total | 80.34 | 111.54 | 40.22 | 42.25 | 57.95 |

five cases from three physical scenarios, including mechanics, electricity, and thermology, while the baseline fails in the last two scenarios due to incorrect message-passing flow because their designed message-passing flow is designed explicitly for Newton-force interaction in the simple mechanical scenario and not flexible enough for other scenarios.

We design two cases in mechanics scenarios: calculating the acceleration and the relative acceleration in the center-of-mass frame. SymDL is designed for handling formula discovery in the simple case, with a specified message-passing flow. Comparatively, our method moves forward to a more general and challenging setting without specifying a message-passing flow representing the formula skeleton. To ascertain the correctness of the learned formulas, besides the baseline, we further introduced a variant of our method without searching the message-passing flow, i.e., directly using the full message-passing block in the first stage. As listed in Figure 4, our model achieved the same performance with two baselines in the simple mechanics case. In the complex case, our model outperformed the SOTA baseline and the variant of our model by a large margin w.r.t. $R^2$ metric. Specifically, these two competitors both failed and got a rather low $R^2$, while our method rediscovered the correct formula with $R^2 = 0.917$, indicating the advantage of searching for the correct messaging-passing flow. For our model, the difference between two formulas for two cases is about the latter two terms corresponding to the additional message-passing flows $\mathbf{V} \to \mathbf{V}'$ and $\mathbf{V} \to \mathbf{u}' \to \mathbf{V}'$, which SymDL cannot handle. The formulas learned by baselines are wrong for lack of necessary dependencies, which fail to have physical meaning and differ largely from the ground truth.

Our problem differs from SymDL, requiring prior knowledge of the formula skeleton for designing the deep learning architecture, which is almost impossible to know in new real-world scenarios. For the rest three cases that the SOTA baseline cannot handle, as listed in Figure 4, the performance gain over the variant using full message-passing flow indicates that optimizing Pareto-optimal score is essential in obtaining correct formulas, which is less subject to redundant message-passing flows that hinder the subsequent SR process, including unnecessary inputs and redundant computation steps. The detailed searching process of the electricity case is analyzed in Section 3.1. For the complex case of thermology, it can be observed that the learned formula successfully captures the effect of externally conducted heat compared with the simple case, while other baselines fail to have physical meanings due to unnecessary inputs and redundant computation steps. Although the change is slight (whether external heat conduction exists), the skeleton and the entire formula are quite different, and so is the entire formula.

To be more precise, the learned message-passing flows by our model are shown in Figure 5. Besides, the time cost of each independent part is shown in Table 2, where we can observe that searching message-passing flows only takes a small part of the whole procedure, and our model's running times are similar to SymDL and usually shorter than FullGN. Furthermore, we conduct experiments to demonstrate the design philosophy of our method, which is reported in Appendix A.9.

**Qualitative Results for Understanding the Searching Process.** We show the searching process of message-passing flow in Figure 6, where the upper row shows the learning curve in terms of error (RMSE), complexity, and a score, which is a weighted summation of error and complexity. From Figure 6, we observe that if the message-passing flow is a sub-structure of the ground-truth message-passing flow, the performance will drop significantly. On the other hand, message-passing flows with redundant layers/inputs/embedding sizes will have similar performance, echoing the rationale of our pruning strategy. The core idea is to search for the most compact message-passing that is expressive enough, and the four-step searching process is as follows, (i) the model tried the number of blocks as $1 \sim 3$, finding similar errors with a rise in complexity, so it opted for 1; (ii) from the searched message-passing flow at the previous stage, it tried to delete every layer associated with $\phi$. It turned out that deleting the edge layer would cause a huge error increase, so the edge layer was preserved, after which it tried to delete the node layer and found that the score decreased (error does not change a lot and the complexity decrease), so it decided to delete node layer; (iii) like the previous stage, it tried to delete each input and found that only deleting the $V \to \phi^u$ connection

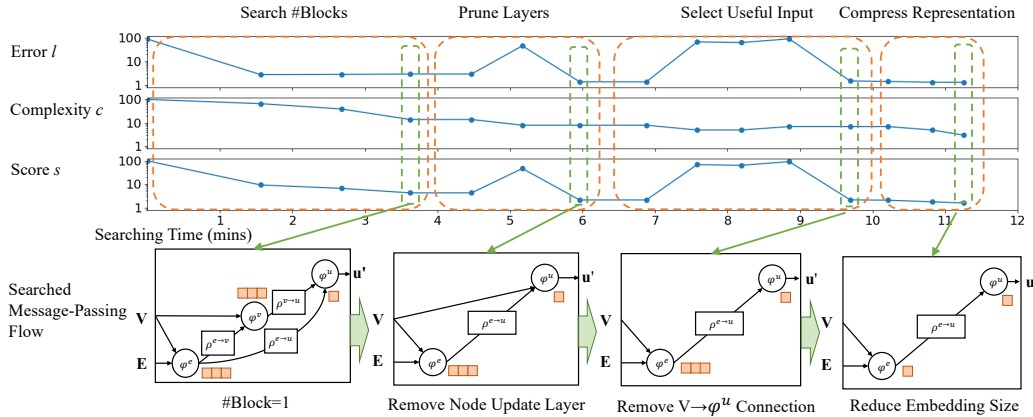

Figure 6: The searching process of message-passing flows in four steps in the circuit scenario.

would not cause an error increase, so this connection was deleted. (iv) finally, it tried to compress each representation and found that when the embedding size was 1, the score minimized, so 1 was chosen as the embedding size. After the whole process, the message-passing flows, including embedding (intermediate variable), functions, and topology, have explicit physical meanings, paving the way for symbolic regression.

## 3.2 Experiments on Real-world Scenarios of Pedestrian Dynamics

To better show how our model discovers unknown graph-structured physical mechanisms in the real world, we conduct carefully-designed experiments of our model on formula discovery for pedestrian dynamics.

**Problem Formulation.** We aim to find a formula that can approximately describe the relationship between acceleration $a$ and the velocity $v$, pedestrian position $x$, and destination position $x_{dest}$. The graph $\mathcal{G}$ describe the interaction relationship, which is constructed as follows when the two pedestrians' distance is less than $R$, they are connected, and otherwise, they are unconnected. Formally, the problem can be described as finding a formula $\mathcal{F}$ that fits $a = \mathcal{F}(\mathcal{G}, x, v, x_{dest})$.

**Datasets.** We conduct experiments on two real-world datasets of crowd trajectories: several experimental datasets from studies about pedestrian dynamics (Boltes & Seyfried, 2013)[1], including the following scenarios, (i) **Unidirectional flow in corridor**: a group of people goes through a corridor in the same direction, as shown in Figure 8(a); (ii) **Bidirectional flow in corridor**: a group of people goes through a corridor in opposite directions, as shown in Figure 8(b).

**Comparing Models.** For pedestrian dynamics, a well-known manually-designed model is the social force model (Helbing & Molnar, 1995), in which pedestrians are with two forces: an attractive force from the destination and a repulsive force from the surrounding pedestrians and obstacles (refer to Appendix A.4 for details).

**Learned Formulas.** The learned formulas and the corresponding physical meanings are reported in Figure 7, which demonstrate that our model can learn different skeletons and formulas that are more precise than the social force model with explicit physical meanings. The performance comparison is also reported in Figure 7, where we can observe that our model has about 10% improvement compared to the social force model.

## 4 Related Works

**Symbolic Regression (SR).** Distilling elegant symbolic expressions from vast experimental data has always been the mainstream method used for finding new formulas and verifying hypotheses throughout the history of physics. SR is a classic topic (Schmidt & Lipson, 2009; Petersen et al., 2020; Biggio et al., 2021; Guimerà et al., 2020) that tries to emulate the process to learn an explicit symbolic model that can describe a projection from input $X$ to the output $y$ as accurately as possible while maintaining its compactness. Traditional methods of discovering formulas from data are primarily based on genetic programming(GP) (Schmidt & Lipson, 2009; Koza, 1994; Worm & Chiu, 2013).

---

[1]https://ped.fz-juelich.de/database/doku.php

| Scenario | Unidirectional flow in corridor | Bidirectional flow in corridor |
|---|---|---|
| Searched Message-passing Flow | | |
| Searched Skeleton | $\varphi^v\left(x_i, \sum_{j\in N(i)} \varphi^e(x_i, x_j), \sum_{j\in V} \varphi^u(x_j)\right)$ | $\varphi^v\left(x_i, \sum_{j\in N(i)} \varphi^e(x_i, x_j)\right)$ |
| Learned Formula (Difference with Social Force) | $\varphi^e$: $ke^{-\lambda r}$ sign($v_\mathbf{n}$) (normal component)
$\varphi^e$: $\delta$ (tangential component)
$\varphi^u$: $v_{mean} = \text{mean}(v_i)$
$\varphi^v$: $\sum_{j\in N(i)} \varphi^e + (v - \lambda_1 v_0 - \lambda_2 \varphi^u)/\tau$ | $\varphi^e$: $k_1\, e^{-\lambda r}$sign($v_\mathbf{n}$) (normal component)
$\varphi^e$: $\delta$ (tangential component)
$\varphi^v$: $\sum_{j\in N(i)} \varphi^e(x_i, x_j) + (v - \lambda v_0)/\tau$ |
| Physical Meanings | 1. There is no repulsive force when the front is faster than the back
2. Tend to dodge to the right
3. Herding: Tend to follow the average speed of the crowd | 1. There is no repulsive force when the front is faster than the back
2. Tend to dodge to the right |
| Performance (R2) Improvements | 0.71 (Social Force Model) $\rightarrow$ 0.78 (Ours) | 0.68 (Social Force Model) $\rightarrow$ 0.81 (Ours) |

Figure 7: Learned formulas for pedestrian dynamics in two scenarios.

Hitherto, there have been promising results yielded by GP-based SR methods such as Burlacu et al. (2020), Virgolin et al. (2019), and the famous commercial SR method Eureqa (Dubčáková, 2011), etc. More recently, methods based on DL (Zheng et al., 2021; Qian et al., 2021; Martius & Lampert, 2016; Kusner et al., 2017; Udrescu & Tegmark, 2020; Udrescu et al., 2020; Daniele et al., 2022) for symbolic regression are introduced with better expressive ability than GP. Furthermore, Cranmer et al. (2020) first proposed to learn graph-structured physical mechanisms (especially kinematics) given formula skeletons. Beyond that, we propose searching for formula skeletons automatically, where existing SR methods can be exploited to look for basic components of the whole formula.

**Graph Neural Network (GNN).** GNN (Kipf & Welling, 2017; Veličković et al., 2018; Gilmer et al., 2017) can be viewed in a message-passing manner (Battaglia et al., 2018; Veličković, 2022; Bronstein et al., 2017), and most of them can be summarized as message-passing among three levels: edge/node/graph level, while the message-passing flows and message/aggregation functions can be customized very differently based on the specific characteristics of applications. It has been widely used for physical systems by capturing the interacting mechanisms, such as simulating mechanical system (Sanchez-Gonzalez et al., 2020; Huang et al., 2021; Sanchez-Gonzalez et al., 2018), designing circuits (Zhang et al., 2019; Ren et al., 2020), simulating heat conduction (Chamberlain et al., 2021; Xhonneux et al., 2020), simulating pedestrian dynamics (Shi et al., 2023; Zhang et al., 2022). Furthermore, there are some works (You et al., 2020; Yoon et al., 2020; Cai et al., 2021; Gu et al., 2021) that adopt automated machine learning techniques for searching the best GNN architecture for a specific prediction task. Unlike them, we focus on the SR problems on graphs and inspired by symbolic regression (Udrescu & Tegmark, 2020), we propose to search the Pareto-optimal message-passing flows, which is both accurate and simple and can benefit learning of symbolic models.

**Pareto-optimal Search.** The previous Pareto-optimal solutions proposed in Neural Architecture Search (NAS) area (Lomurno et al., 2021; Lu et al., 2020; Dong et al., 2018) focus on finding the model architecture with both high prediction accuracy and low inference latency, which does not meet requirements for solving graph SR problem. Instead, our proposed method is based on a novel insight in the SR scenario: the performance would be similar when the message-passing flow (skeleton) is a super-structure of the ground-truth one. In contrast, the performance degrades a lot if it is a sub-structure of the ground-truth one.

## 5 CONCLUSION

In this paper, we generalize the problem in Cranmer et al. (2020) by learning the formula skeleton rather than manually designing, which is crucial for learning formulas in a new physical area without much prior knowledge. We propose a new SR method that first transforms the discovery of the formula skeleton to the search of the Pareto-optimal message-passing flow with accuracy and compactness and then symbolizes its message functions to obtain the underneath formula. We conduct experiments on five datasets from three different physical domains, including mechanics, electricity, and thermology, demonstrating that our method can consistently learn plausible formulas describing the graph-structured physical mechanism. Furthermore, to show that our model is practical for learning unknown formulas in the real world, we conduct experiments on two real-world datasets about pedestrian dynamics, which learn different formulas with explicit physical meanings for different scenarios more precisely than mainstream empirical formulas.

## ACKNOWLEDGEMENT

This work was supported in part by the National Key Research and Development Program of China under 2020YFA0711403, the National Nature Science Foundation of China under 61971267, U1936217, 61972223, 62171260. Q. Yao was in part supported by NSFC (No. 92270106) and CCF-Baidu Open Fund.

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

## A EXPERIMENTS

### A.1 DATA GENERATION

Besides the scenarios of mechanics and electricity, we further illustrate the scenario of thermology as follows.

**Example 4 (Thermology: Heat Conduction)** *The objective of this problem is to compute the entropy production rate. The edge update function $\phi^e$ corresponds to Fourier's Law of Heat Conduction, and the node update function $\phi^v$ corresponds to the Clausius entropy expression, followed by an aggregation $\rho^{v \to u}$ that sums up individual entropy production rates.*

In different scenarios, we devised different inputs and computed the theoretical outcome according to the known mechanisms.

In the scenario of *Mechanics*, we randomly (standard normal distribution) set the $(x, y)$ coordinates of the particles in 2-D cases. We assign the masses of the particles randomly according to the lognormal distribution. The original lengths of the springs are all set to 1. In the complex case, external forces with all dimensions following the standard normal distribution are exerted. Graph topology is picked randomly. We then compute the accelerations of each particle with Hooke's Law, the independent action principle of force, and Newton's Second Law of Motion.

In the scenario of *Electricity*, we randomly give a topology on the graph and set the electric potential of nodes following the standard normal distribution. Resistances of resistors on edges are chosen uniformly randomly from 0.01 to 1.01 to avoid extremely large power outputs. We then compute the power of each edge (resistor) according to Joule's Law and add them up to reach the overall power of the resistor circuit.

In the scenario of *Thermology*, the graph topology is given as a 'grid', echoing the core idea of Finite Element Analysis. We randomly set the temperature of each node between 0 and 1 and the thermal conductivity between 1 to 3 globally. We then compute the discrete laplacian on the grid and the heat flow according to Fourier's Law of Heat Conduction. With each node's heat flow and temperature, we compute their entropy production rate separately and add them up to reach the overall entropy production rate.

The basic information of our used datasets is listed in Table 3.

Table 3: Basic Information about Datasets.

| Domain | Mechanics | Electricity | Thermology |
|---|---|---|---|
| Scenario | Spring-connected particles | Resistor circuit | Thermal system |
| Topology | Unoriented graph | Unoriented graph | Grid |
| Node Input | Mass, position, (force) | Electric potential | Temperature, (external heat) |
| Edge Input | Spring constant | Resistance | Thermal conductivity |
| Global Input | None | None | None |
| Output | Acceleration (relative to mass center) | Overall power | Overall entropy production |
| # Graphs | 100 | 100 | 50 |
| # Nodes | 5 | 5 | 9 |
| # X-Y pairs | 100000 | 100000 | 100000 |

### A.2 REPRESENTATIVE SNAPSHOTS OF PEDESTRIAN DATASETS

To better understand the pedestrian scenarios, we show two representative snapshots of two pedestrian datasets in Figure 8: unidirectional flow in a corridor and bidirectional flow in a corridor.

### A.3 BASELINE DETAILS

The message-passing flows of baselines, SymDL, and FullGN are shown in Figure 9, and their applicability in different scenarios are demonstrated in Table 4.

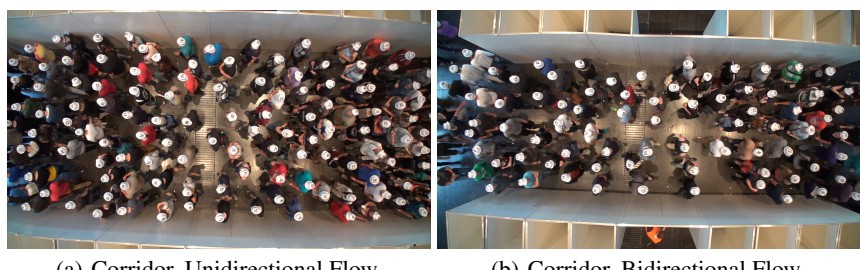

(a) Corridor, Unidirectional Flow.          (b) Corridor, Bidirectional Flow.

Figure 8: Representative snapshots of two pedestrian datasets.

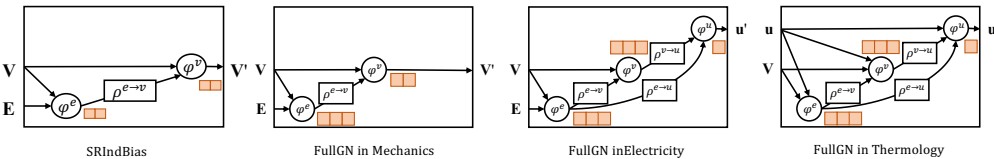

Figure 9: message-passing flows of SymDL and FullGN.

### A.4 DETAILS OF SOCIAL FORCE MODEL

In the social force model, the baseline model for pedestrian scenarios, the dynamics of pedestrians are driven by two factors: (a) a pedestrian is attracted by his/her destination with force $\mathbf{F}_{Di} = (v_{di}\mathbf{e}_i - \mathbf{v}_i)/\tau$, $\mathbf{e}_i = \frac{\mathbf{x}_d - \mathbf{x}_i}{\|\mathbf{x}_d - \mathbf{x}_i\|}$, where $\mathbf{v}_{di}$ is the value of desired velocity, $\mathbf{v}_i$ is the current velocity, $\tau$ is the relaxation time, $\mathbf{e}_i$ is the unit vector toward the destination; (b) the nearby ones repulse a pedestrian with a force $\mathbf{F}_{ij} = A_i \exp\left(-r_{ij}/B_i\right)\mathbf{e}_{ij}^n$, where $\mathbf{F}_{ij}$ is the repulsive force, $r_{ij}$ is the distance between pedestrian $i$ and $j$. The joint force is $\mathbf{F}_i = \mathbf{F}_{Di} + \sum_{j\in\mathcal{N}_i}\mathbf{F}_{ij}$, where $\mathcal{N}_i$ means the set of pedestrians whose distance to pedestrian $i$ is less than 5 meters. The social force model is widely used as the foundation of much commercial software such as viswalk[2] and anylogic[3]. In this paper, we assume that the mass of a pedestrian is 1, and thus $a_i = F_i/m = F_i$. However, on the one hand, the social force model is manually designed, which may have discrepancies with real-world pedestrian dynamics. On the other hand, different scenarios usually have very different pedestrian interaction mechanisms, which one single model cannot precisely model. So it is meaningful to learn data-driven formulas to describe different pedestrian interaction mechanisms.

### A.5 IMPLEMENTATION

We implement our model in Python using Pytorch library and optimize all the models by Adam optimizer (Kingma & Ba, 2015). We use parallel symbolic regression in Python (PySR)[4] (Cranmer, 2020) to extract formulas from each message functions $\phi$.

### A.6 PARAMETER SETTINGS

For the DL part, we set the learning rate as $10^{-4}$, tolerance in early stopping as 10, #layers and embedding size in MLP as 4 and 16, the max number of epochs as 20000 and the weight $\lambda$ as 0.1. The choice of parameter $\lambda$ is analyzed in Appendix A.8. For the traditional SR part, our candidate symbols include both binary operator $\{+, -, \times, /\}$ and unary operator $\{\text{sign}, \exp\}$, and we set batch size as 1024.

---

[2]https://www.myptv.com/en/mobility-software/pedestrian-simulation-software-ptv-viswalk
[3]https://www.anylogic.com/features/libraries/pedestrian-library/
[4]https://github.com/MilesCranmer/PySR

Table 4: The applicability in different scenarios.

| Method | Mechanics (simple) | (complex) | Electricity | Thermology (simple) | (complex) |
|--------|--------------------|-----------|-------------|---------------------|-----------|
| SymDL  | √                  | √         | ×           | ×                   | ×         |
| Ours   | √                  | √         | √           | √                   | √         |

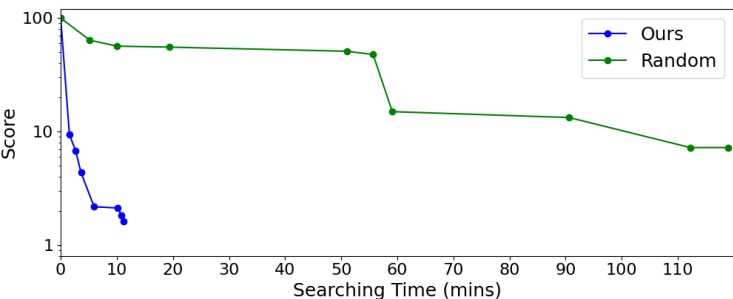

Figure 10: The searched best score v.s. searching time for two different searching methods in the circuit scenario.

## A.7 COMPARING HIERARCHICAL PRUNING WITH RANDOM SEARCH

Furthermore, to demonstrate the effectiveness of our search method, we compare it with the random searching strategy and plot their searching processes in Figure 10. From that, we can observe that our method is much more efficient than the random search algorithm, which suggests that the searching problem is difficult and that our method effectively reduces the original colossal search space. Specifically, even if the random search algorithm takes ten times longer than ours, the score of the best-searched skeleton is still 4.5 times worse than ours, and the searched skeleton is wrong.

## A.8 PARAMETER SENSITIVITY

One of the most critical parameters in our model is the weight $\lambda$ that balances the complexity and errors. We normalize the input-output pairs to make the outputs have a standard deviation of 1. Specifically, for each dimension of features, we divide the features by their standard deviation, maintaining the fitting errors with similar magnitudes. Since the complexities of different skeletons are also with similar magnitudes, the best value of $\lambda$ is similar for different datasets. As shown in Table 5, we test different values of $\lambda$ on three diverse scenarios, including the circuit scenario and two mechanical scenarios. For each value of $\lambda$, we use ten different seeds to train the model to test whether our model can learn the correct message-passing flows and the correct formulas. We choose the best formula among ten formulas, so all these values of $\lambda$ can allow us to find the correct formula in different scenarios. Among them, we choose $\lambda = 0.1$ for achieving the highest success rate among all three scenarios.

## A.9 DEMONSTRATION OF DESIGN PHILOSOPHY

To demonstrate the design philosophy of our method, we plot several learning curves of different message-passing flows in the simple mechanical scenario.

We first plot the learning curves of three different message-passing flows in Figure 11, corresponding to the one lacking the necessary message-passing connection, the ground-truth one, and the one with a redundant message-passing connection, respectively. We find that the ground-truth message-passing flow and the one with redundant message-passing connection have only little variations (the RMSE is around 0.1 and the MAPE is around 1.0) in performances after the loss function converges. However, the performance of message-passing flow lacking necessary connections decreases significantly (the RMSE is about 1.4, and the MAPE is 4.0).

We further test the impact of the embedding size on the performance, where the learning curves are shown in Figure 12. From that, we can see that when the embedding size is less than a certain number

Table 5: Success Rate in the Different Scenarios.

| Value of $\lambda$ | | 0.025 | 0.05 | 0.1 | 0.2 | 0.4 |
|---|---|---|---|---|---|---|
| Circuit | learning message-passing flows | 3/10 | 9/10 | 10/10 | 10/10 | 8/10 |
| | learning formulas | 3/10 | 9/10 | 10/10 | 10/10 | 8/10 |
| Mechanics (Simple) | learning message-passing flows | 2/10 | 8/10 | 9/10 | 6/10 | 1/10 |
| | learning formulas | 2/10 | 7/10 | 8/10 | 6/10 | 1/10 |
| Mechanics (Complex) | learning message-passing flows | 1/10 | 6/10 | 7/10 | 5/10 | 2/10 |
| | learning formulas | 1/10 | 5/10 | 6/10 | 5/10 | 1/10 |

Figure 11: Learning curves w.r.t. message-passing flows.

2, the performance decreases significantly (RMSE is more than 1.4), while the performance is similar (RMSE is around 0.2) when the embedding size is not less than 2.

Last, we test whether the softmax function can learn the ground-truth aggregator. As we can see in Figure 13, the learning curve with softmax and with the ground-truth aggregator is quite similar, and we verify that the learned aggregator is the same as the ground truth.

# B  METHOD DETAILS

## B.1  USAGE OF THE PROPOSED METHOD

The result obtained by our method is fairly stable in terms of both searching formula skeleton and symbolizing learned neural networks. First, our designed searching method can guarantee that only better graph structures (message-passing flows) get selected during the learning process. Based on an insightful observation that the loss would increase a lot when the graph structure is a subset instead of a superset, we propose to search four components of graph structure (block, layer, connection, dimension) by starting with a full structure and then pruning to obtain a compact one. Second, the stability of SR results has also been demonstrated by applications in Cranmer et al. (2020).

When applied to a new physical domain, we can directly use the same DL-related hyper-parameters (such as the learning rate and embedding size of MLP) on old domains and slightly tune the search-related hyper-parameter (weight $\lambda$) near the previous optimal value on old domains. The stability of the proper $\lambda$ value is demonstrated in Table 5. Due to the randomness of DL training and the

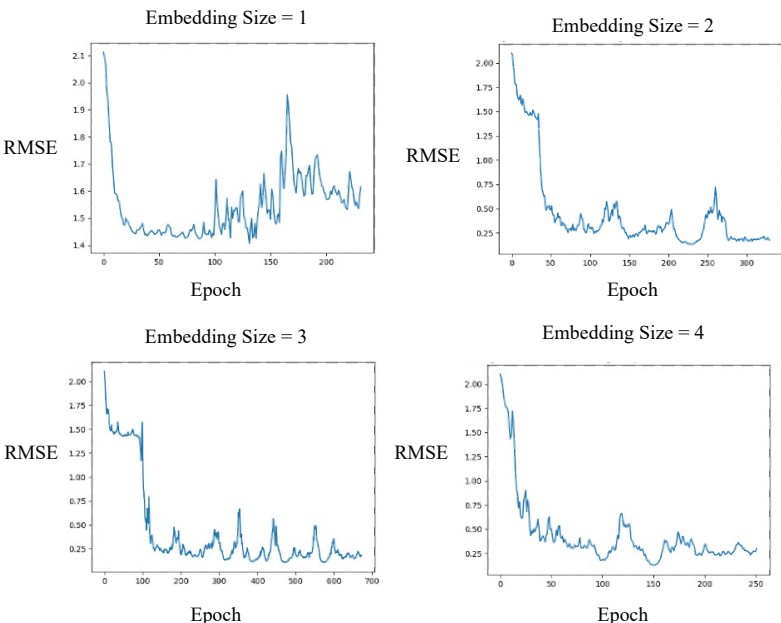

Figure 12: Learning curves w.r.t. embedding sizes.

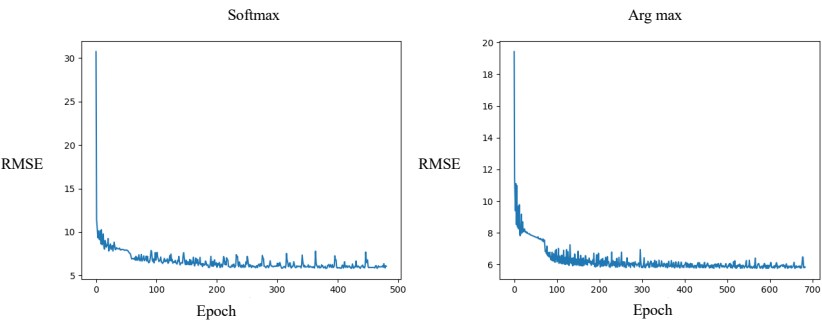

Figure 13: Learning curves with or without softmax.

genetic algorithm, we train a model ten times with different seeds to get ten formulas and select the Pareto-optimal formula according to the score (see Appendix B.6) as the final formula. Overall, the above process in a new domain does not require heavy human labor.

## B.2 TRAINING ALGORITHM

The training algorithm of our model is summarized in Algorithm 1.

## B.3 DETAILS OF SYMBOLIZING THE AGGREGATION/MESSAGE FUNCTIONS

**Symbolize Aggregation Function** $\rho$. We choose several commonly used aggregators as candidates, including *sum*, *mean* and *max*, while other candidate aggregators such as *min* can be achieved by $min(x) = -max(-x)$, and root mean square can be represented by $(mean(x^2))^{\frac{1}{2}}$. Harmonic mean and l-norm can also be decomposed into two $\phi$ functions and a *mean* aggregators. These operations can be learned by $\phi$.

---

**Algorithm 1 The process of obtaining the graph-structured symbolic model**.

---

**Require:** $\mathcal{D}$ including training data $\{(X, Y, \mathcal{G})\}$;
1: (start stage 1) search message-passing flow follows Section 2.3;
2:     step 1: search the Pareto-optimal block number;
3:     step 2: search the required layers;
4:     step 3: search the necessary input variables;
5:     step 4: search the Pareto-optimal embedding sizes;
6:     train the model with Pareto-optimal message-passing networks;
7: (start stage 2) Symbolize Aggregation Function $\rho$ and $\phi$
8:     replace each $\rho$ by the aggregator with largest weight;
9:     train the model and record the input-output pairs of each $\phi$;
10:     replace each $\phi$ by a formula obtained by classic SR, with constants left to be fitted on the data;
11:     fit the correct constants from the data by gradient descent and get the final graph-structured symbolic model.

---

**Symbolize Message Function $\phi$.** After training the GNN model, we use the tool of symbolic regression to extract a symbolic model with correspondence to message and update functions. Specifically, for message functions in the neural network, we record their input-output pairs and implement classic symbolic tools to symbolize them.

**Retrain and Fine-tune Constants.** Finally, to eliminate accumulated errors, we set all constants in the entire formulas as parameters to be trained and fine-tune them to get the final graph-structured mechanism represented by symbolic models. Specifically, after extracting the message-passing flows, we replace the MLP in message/update functions with the corresponding formulas. And we set the constants in the formulas as parameters in the deep model to optimize (all the operations in symbolic models are differentiable).

### B.4    DETAILS OF ERRORS

In this paper, we use RMSE as the error measure. RMSE can be calculated as

$$RMSE = \sqrt{\frac{\sum_{i=1}^{n} (y_i - \hat{y}_i)^2}{n}},$$

where $\hat{y}_i$ is the $i$-th predicted value and $y_i$ is the $i$-th ground-truth value and $i = 1, \cdots, n$.

### B.5    DETAILS OF COMPLEXITY

The design of complexity measurement for message-passing flows is flexible. In this paper, we calculate the complexity as follows, (i) for each layer, the complexity can be calculated as the product of the embedding size of this layer and the number of inputs in this layer; (ii) the whole complexity of the message-passing flow can be calculated as the summation of the complexity of each layer. To illustrate, the complexity of four message-passing flows in the Figure 6 can be calculated as $2 \times 3 + 2 \times 3 + 2 \times 1 = 14, 2 \times 3 + 2 \times 1 = 8, 2 \times 3 + 1 \times 1 = 7$ and $2 \times 1 + 1 \times 1 = 3$. As we can see, the complexity consistently decreases in the search process.

### B.6    DETAILS OF SCORE

The score of a graph-structured formula is $s = l + \lambda c$, where $l$ is RMSE, $c$ is the complexity of the graph-structured formulas, which is defined as the complexity of message-passing flow multiplying the average complexity of component formulas.

## C    DETAILED RELATED WORKS ON SR

Martius & Lampert (2016) proposed a model named EQL and extracted the symbolic formulas with a neural network with symbolic models as building blocks. Kusner et al. (2017) managed to eschew the problem of discrete optimization by converting discrete data into a parse tree. AI Feynman (Udrescu

& Tegmark, 2020; Udrescu et al., 2020) split the function into sub-functions and performed regression on each module separately. The partition of functions was achieved with a trained neural network. SymDL (Cranmer et al., 2020) exploited the inductive biases of the correspondence of physical mechanisms (kinematics, especially) to GNN structure and established the method PySR to tackle the problem. They first revealed the link between GNNs and physical mechanisms. Unlike these works, our model can learn graph-structured physical mechanisms without requiring the information of formula skeletons, which can hardly be obtained in new physical scenarios for discovery.

