# OpenReview forum: "Learning Symbolic Models for Graph-structured Physical Mechanism"
_ICLR.cc/2023/Conference — ICLR 2023 poster_

### Official Review · Reviewer_H8ss · 2022-10-24

**Confidence:** 2
**Correctness:** 3
**Technical Novelty And Significance:** 3
**Empirical Novelty And Significance:** 3
**Recommendation:** 5

**Clarity, Quality, Novelty And Reproducibility:**

Quality: average. Text could be more polished.
Clarity: could be improved.
Originality: the work seems to be original, specially mixing SR with GNN.

**Strength And Weaknesses:**

+ the idea of formulating the problem as a two step optimization problem, although not new, seems good in the context of this work
- writing is a bit difficult to follow with some typos and sentences difficult to understand.

**Summary Of The Paper:**

This paper presents an approach that generalizes symbolic regression
to graph-structured physical mechanisms. As opposed to classical
Symbolic Regression, this work assumes that X and Y in y=F(x) can be
both represented as graphs. The method is based on a two-level
optimization procedure where first the formula skeleton is modeled
with a message-passing flow and parameters are learned. In a second
step, symbolization is used to each Deep Learning component.


**Summary Of The Review:**

This paper presents an approach that generalizes symbolic regression
to graph-structured physical mechanisms. As opposed to classical
Symbolic Regression, this work assumes that X and Y in y=F(x) can be
both represented as graphs. The method is based on a two-level
optimization procedure where first the formula skeleton is modeled
with a message-passing flow and parameters are learned. In a second
step, symbolization is used to each Deep Learning component.

Questions, comments
-------------------

s2.3, p4: for each layer: what do you mean by layer (in a general
graph)? Why do you need layers?

For the simple case in mechanics scenario The detailed --> ??

Step 1: predicted acceleration: shouldn't the model be generic?

Obtaining the final formula: you haven't mentioned genetic algorithms
in the methodology. What are the parameters? (population size,
population format, fitness function etc)

Is it fair to compare your model with SymDL, since their purposes and
knowledge representation are different?

There are some logic-based systems that can also learn formulas and
functions. Why aren't they included in the related work and why aren't
they compared against your work? (e.g. https://arxiv.org/abs/2208.11561).


Other comments, typos etc
-------------------------

following bi-level the optimization --> following the bi-level optimization
in this layers --> in this layer (in these layers?) (2x)

cannot by handled --> cannot be handled

can describe a the projection --> ??

---

> ### Author Response · Authors · 2022-11-19
> **Improve the clarity and thoroughly revised our paper.**
>
> Thanks for your helpful and valuable comments! We very much appreciate it. Please allow us to respond to your questions one by one as follows.
>
> **Q1**: s2.3, p4: for each layer: what do you mean by layer (in a general graph)? Why do you need layers?
>
> **Re**: Thanks for your careful review. A layer means one of the update functions ($\phi_u$, $\phi_v$, $\phi_e$), which is corresponding to a component formula in the context of SR. We need these layers for compose the complete formulas. We have revised the paper by formally defineing  this term.
>
> **Q2**: For the simple case in mechanics scenario The detailed --> ??
>
> **Re**: Thanks for your careful reviewing. We are sorry for this typo. Actually this sentence should be removed and instead we describe the simple case in mechanics scenario in Appendix A.1.
>
> **Q3**: Step 1: predicted acceleration: shouldn't the model be generic?
>
> **Re**: Thanks for your careful reviewing. We agree with you and have changed it to "predicted values".
>
> **Q4**: Obtaining the final formula: you haven't mentioned genetic algorithms in the methodology. What are the parameters? (population size, population format, fitness function etc)
>
> **Re**: Thank you for your valuable questions. For transforming the DL component function, all GP and DL-based methods proposed for classic SR can be used in this part. In our implementation, to transform the learned DL component function into (sub-)formulas, we directly utilized PySR (a package of SR) with default settings for learning component formulas and use the default settings, please refer to https://github.com/MilesCranmer/PySR for its details. In terms of customized parameters for this part, our candidate symbols include both binary operator $\{ +, -, \times, / \}$ and unary operator $\{ \text{sign}, \text{exp}\}$, and we set batch size as $1024$. We introduced the detailed parameter settings in Appendix A.6.
>
> **Q5**: Is it fair to compare your model with SymDL, since their purposes and knowledge representation are different?
>
> **Re**: Thanks for your valuable question. We agree with you that our method and SymDL are with different purposes and knowledge representation. In fact, our method generalize SymDL by automatically learning formula skeletons from the data instead of mannual design according to prior knowledge. We compare our method with SymDL to demonstrate that in cases where the prior knowledge adopted by SymDL is correct, i.e., the simple mechanics case, two methods achieve similar performance; while in another complex case our method significantly outperforms due to capability of automatic skeletons learning.
>
>
> **Q6**: There are some logic-based systems that can also learn formulas and functions. Why aren't they included in the related work and why aren't they compared against your work? (e.g. https://arxiv.org/abs/2208.11561).
>
> **Re**: We propose a general framework for learning symbolic model for graph-structured mechanisms, which is very different from the traditional SR settings, where your mentioned work falls in. In our framework, existing SR methods including your mentioned one can be exploited as a sub-solver to obtain basic symbolic components of the whole formula (in this paper we use PySR). In this new version, we have discussed your mentioned paper in the related work.
>
>
> **Q7**: Other comments, typos etc:
> following bi-level the optimization --> following the bi-level optimization in this layers --> in this layer (in these layers?) (2x)
> cannot by handled --> cannot be handled
> can describe a the projection --> ??
>
> **Re**:  Thank you for the careful review. We have corrected your mentioned typos and thoroughly proofread the paper.

---

> ### Author Response · Authors · 2022-12-09
> **We are willing to have a further discussion, especially about the novelty of our work and the clarity of our paper**
>
> Dear reviewer,
>
> The discussion period is almost ending. Could you please confirm whether our responses have alleviated your concerns? If you have further comments, we will also be happy to discuss them.
>
> We have improved the clarity and thoroughly revised our paper for your interest. Also, in the general comments, we highlight the novelty, i.e., the difference between our work and SymDL, and insights from SR inspiring us to design our method.
>
> Thank you very much!

---

### Official Review · Reviewer_dEBy · 2022-10-25

**Confidence:** 4
**Correctness:** 3
**Technical Novelty And Significance:** 2
**Empirical Novelty And Significance:** 2
**Recommendation:** 6

**Clarity, Quality, Novelty And Reproducibility:**

Questions/comments about clarity:

1. I do not understand why there are multiple numbers of blocks and how they lead to final formula. From my understanding, if \phi_e occurs in different blocks, the final formula will also contain two \phi_e operations, but it seems the learned formulas in Figure4 are quite concise and do not have such overlaps.

2. Definition 2: what is V', E'? Do you mean y is from a different graph? From my understanding y should be from the same graph but does not have overlap with X_i, i.e. y_i \in {V, E, u}/X_i.

3. typo: “the following bi-level the optimization” -->"the following bi-level optimization"

**Strength And Weaknesses:**

Strengths:
1. It provides a good problem setting. Symbolic regression on graph-structured data is an interesting topic.
2. The proposed method is generally intuitive and reasonable. It is easy to understand that not all flows are useful and pruning the flows to get a compact one is a good way to obtain a concise formula.
3. Nice visulaization and good experimental results.

Weaknesses:
1. The methodology of this paper is basically following (Cranmer et al. 2020). Its generalization seems essentially a combination of NAS and Symbolic Regression. It is useful, but slightly overstated. For most of the physical systems, we know the mechanism and the design of the "flow" seems definite and there is no need to search it. The authors showed one case in Appendix A.4, but it is still not so attractive, because the search space of the flows is not large. The choices of reasonable flows are actually limited.
2. The definition of complexity and its weight may largely impact the results. The model is unstable. We may need to run multiple times to get the best formula.
3. Presentation can be improved. For example, A.4 is important to demonstrate the advantage of the method (as described in the conclusion), I believe it should be put in main body of the paper instead of Appendix. The Figure 3 is described in a wierd place. In fact, in Sec 2.3 it only described the Fig 3(a), but not the whole framework. There are also some other unclear points as below.

**Summary Of The Paper:**


The paper studies the problem of symbolic regression given the graph structure. It generalizes a previous approach (Cranmer et al. 2020) by additionally learning the formula skeleton. The method is seperated into two steps: searching the Pareto-optimal message passing flows and component-wise symbolic regression. Compared to (Cranmer et al. 2020), the second stage is not new, and the main novelty is to learn and prune the formula skeleton rather than manually designing one. Experiments on different scenarios demonstrate that the proposed method can learn better formulations and it has better prediction performance.

**Summary Of The Review:**

The paper is solving an interesting problem and it provides a decent solution. However, the significance and practicality are not convincing. The clarity also needs some improvement.

---

> ### Author Response · Authors · 2022-11-19
> **Significance and practicality. (2/2)**
>
>
> **Q3.1**: Presentation can be improved. For example, A.4 is important to demonstrate the advantage of the method (as described in the conclusion), I believe it should be put in main body of the paper instead of Appendix.
>
> **Re**: We have moved Appendix A.4 into the main text, and shortened the section of related works by moving some details into Appendix C.
>
> **Q3.2**: The Figure 3 is described in a wierd place. In fact, in Sec 2.3 it only described the Fig 3(a), but not the whole framework.
>
> **Re**: Thank you for you helpful comment. Based on your comment, we adjust the placement of the Fig 3 and we describe Fig 3(a) in Sec 2.3 and describe Fig 3(b) in Sec 2.4.
>
> **Q4**: I do not understand why there are multiple numbers of blocks and how they lead to final formula. From my understanding, if \phi_e occurs in different blocks, the final formula will also contain two \phi_e operations, but it seems the learned formulas in Figure4 are quite concise and do not have such overlaps.
>
> **Re**: Thank you for this valuable comment. For the complex case in mechanics scenarios, we need two blocks to be a superstucture of the ground-truth message-passing flow for pruning. For this case, the ground-truth layers are edge→global→node, and since each block consist edge→node→global (the order is different), which is not a super-stucture of the ground-truth flow, we need to cascade two blocks and then prune some layers as following,
> edge→node(prune)→global→edge(prune)→node→global(unused),
> and obtain edge→global→node.
>
>
> **Q5**: Definition 2: what is V', E'? Do you mean y is from a different graph? From my understanding y should be from the same graph but does not have overlap with $X_i$, i.e. $y_i \in \{V, E, u\}/X_i$.
>
> **Re**: They are two sets of variable (known variables and unknown variables) on the same graph $\mathcal{G}$ and do not have overlap. $\{V, E, u\}/X_i$ is slightly different from our meaning, such as in mechanical scenarios, both of the input and output contains node-level variables ($V$).
>
> Based on your valuable question, we improve the description as following:
> Given a set of $\{ (\mathcal{G}_i,X_i, \mathbf{y}_i) \}$, where $X_i\subset \{\mathbf{V}, \mathbf{E}, \mathbf{u}\}$, which are known variables, $\mathbf{y}_i \in \{\mathbf{V}' ,\mathbf{E}',  \mathbf{u}'\}$, which are unknown variables, and variables with prime denotes output variables, we aim to find an accurate and compact formula $\mathcal{F}(\cdot)$ that fits $\mathbf{y} = \mathcal{F}(\mathcal{G},X)$.
>
>
> **Q6**: typo: “the following bi-level the optimization” -->"the following bi-level optimization"
>
> **Re**: Thank you for the careful review. We have corrected your mentioned typos and thoroughly proofread the paper.

---

> > ### Comment · Reviewer_dEBy · 2022-12-10
> > **Thank you for the clarification**
> >
> > The presentation of the paper is improved, and my questions about clarity are also answered well. Regarding the main issue, i.e. novelty and practicality, I appreciate the authors' explanation but do not fully accept it. I still have some concerns about the efficiency and necessity of search skeletons in many scenarios; however, I agree it is useful for pedestrian modeling, and it is good to be parameter-insensitive. So, I increase my score to 6.

---

> ### Author Response · Authors · 2022-11-19
> **Significance and practicality. (1/2)**
>
> Thanks for your helpful and valuable comments! We very much appreciate it. Please allow us to respond to your questions one by one as follows.
>
> **Q1.1**: The methodology of this paper is basically following (Cranmer et al. 2020). Its generalization seems essentially a combination of NAS and Symbolic Regression.
>
> **Re**: Thank you for your valuable comment. You are right that our generalization is based on a combination of NAS and Symbolic Regression. Technically, our NAS method design is significantly different from existing NAS works. Specifically, our hierarchical pruning method is inspired by the unique insight from SR as following:
>
> The performance would be similar when the formula skeleton (message-passing flow) is a super-structure of the ground-truth one, while the performance degrades a lot if it is a sub-structure of the ground-truth one (shown in Figure 11 & 12).
>
> In a word, we design a NAS-based new method for the problem of SR on the graph.
>
> **Q1.2**: It is useful, but slightly overstated. For most of the physical systems, we know the mechanism and the design of the “flow” seems definite and there is no need to search for it. The authors showed one case in Appendix A.4, but it is still not so attractive, because the search space of the flows is not large. The choices of reasonable flows are actually limited.
>
> **Re**: Thank you for this comment. First, learning the skeleton in the unknown scenario is of great importance for pedestrian modeling. For example, in the scenario of pedestrian dynamics, the formula skeletons are various in related works [1,2,3] corresponding to diverse physical meanings, which depend on specific scenarios. Manually designing such formula skeletons is difficult, so learning the skeleton from data is very crucial. Please also refer to the 1st statement in the general comments.
>
> Second, we would like to point out that, even though we know the mechanism, a slight change of the settings can lead to largely different message-passing flows. For example, as shown in Figure 5, in the thermology scenario, the existence of external heat flow can lead to significantly different message-passing flows; in the mechanical scenario, when we change the frame of reference from the natural reference system into the center of mass system, the message-passing flow varies a lot.
>
> Third, we add an experiment in Appendix A.7 to demonstrate the difficulty of searching flows, highlighting the necessity of our designed hierarchical pruning-based method. Specifically, we compare our searching method with the random searching strategy and plot their searching processes in Figure 10. Our method is much more efficient than the random search algorithm, suggesting that the searching problem is difficult and our method effectively reduces the original huge search space. Specifically, even if the random search algorithm takes 10 times longer than ours, the score of the best-searched skeleton is still 4.5 times worse than ours, and the searched skeleton is wrong.
>
> **Q2**: The definition of complexity and its weight may largely impact the results. The model is unstable. We may need to run multiple times to get the best formula.
>
> **Re**: Your mentioned problem about the definition of complexity and its weight exists in most existing SR methods. And many SR methods require running multiple times to select the best formula. We conduct an experiment in Appendix A.8 to show that our method is not parameter-sensitive. As shown in Table 5, if we only run our method once using the best hyper-parameters, the success rate is fairly high; and if we run it 10 times to get rid of the sensitivity of seed[1], our method can stably learn the correct formulas when the $\lambda$ is in (0.025, 0.4).
>
> Furthermore, we have a special design to make it more stable: we normalize the input-output pairs to make the outputs have a standard deviation of $1$. Specifically, for each dimension of features, we divide the features by their standard deviation, and then the errors are in similar magnitudes, and besides, the complexities of different skeletons are also with similar magnitudes, so the best value of $\lambda$ is similar for different datasets. This manipulation will not affect the constants in the formula, so when we retrain the constants, we will use the original input-output pairs.
>
> For better practicality, we add a subsection (Appendix B.2) to describe how to use our model and why it is robust.
>
> [1] Mundhenk T N, Landajuela M, Glatt R, et al. Symbolic regression via neural-guided genetic programming population seeding. NeurIPS, 2021.

---

> ### Author Response · Authors · 2022-12-09
> **We are willing to a have further discussion about the significance and stability issue**
>
> Dear reviewer,
>
> The discussion period is almost ending. Could you please confirm whether our responses have alleviated your concerns? If you have further comments, we will be happy to discuss them.
>
> For your interest, we (1) clarify the difference on problem definition and methodology; (2) explain the stability issue; (3) re-organize the presentation. Please refer to the details in both the responses for you and the general comments.
>
> Thank you very much!

---

### Official Review · Reviewer_oCCc · 2022-10-25

**Confidence:** 4
**Correctness:** 4
**Technical Novelty And Significance:** 4
**Empirical Novelty And Significance:** 3
**Recommendation:** 8

**Clarity, Quality, Novelty And Reproducibility:**

As mentioned in the above review, the paper is generally well-written but still has room for improving clarity. As for its novelty, I like the core idea of transforming the discovery of the formula skeleton into the search for the Pareto-optimal message-passing flow with accuracy and compactness, although the proposed method is built upon the SR method in Cranmer et al. 2020. Still, it is also reasonable and highly insightful. The authors have provided source code for reproducibility check.

**Strength And Weaknesses:**

Strengths:
1. The studied problem is important and interesting. AI for science is a highly promising application field that has not been well explored. This paper has made an essential footstep toward finding scientific formulas in a more general and automatic way.
2. Compared with existing works, this paper has tackled a more practical and challenging problem of learning the formula skeleton rather than manually designing, which is crucial for learning formulas in new physical domains with less prior knowledge.
3. The proposed method is novel, especially the idea of transforming the discovery of the formula skeleton to the search for the Pareto-optimal message-passing flow with accuracy and compactness.
4. The two main designs are technically sound. First, the transformation from learning skeleton to searching message-passing flow is correct as the latter corresponds to explicit meanings in the symbolic calculation for graph-structured mechanisms. Second, the design philosophy behind the proposed pruning-based search procedure is reasonable, based on an insightful observation that when the searched graph structure is a subset of ground truth instead of a superset, the optimized score/loss will increase significantly.
5. The authors conduct extensive experiments. The proposed method can find correct formulas in five mechanisms with different difficulties, while compared methods only succeed in a simple case, i.e., mechanics as in Cranmer et al. 2020. Moreover, it can discover a new analytic formula that can predict real-world pedestrian dynamics more precisely.
6. Overall, The paper is well-written and easy to follow.

Weaknesses:
1. For easy-to-use concerns, the paper should have a specific subsection that provides a practical guide on using the proposed method when discovering formulas in new physical domains. Currently, the related information is scattered in Sec. 3.1, Appendix A.3 and A.4, not concentrated enough.
2. To better demonstrate the superiority of the designed pruning-based search procedure, authors can consider adding further studies in the appendix. For example, how about comparing with a random search method that does not decompose searching steps and does not leverage a pruning strategy? Also, Figure 6 can be improved by plotting four steps together, with the x-axis being absolute training time. This can make the explanation text much easier to follow.
3. The clarity of this paper can be further improved as follows,
    - Use a specific figure to demonstrate the design philosophy of the pruning-based search procedure.
    - Give more details on the normalization process, as in “We normalize the input-output pairs to make the outputs have variance 1” (Appendix A.3).
    - Make related work part shorter and save space for results analysis, especially the real-world example in A.4.
4. A few typos. e.g., on page 8, “the time cost of each independent part is shown in Table 6” should be Table 3.


**Summary Of The Paper:**

This paper improves SymDL by searching the Pareto-optimal message-passing flows to learn an additional formula skeleton. Such improvement empowers the method with more generality (requiring no prior knowledge), compactness, while maintaining correctness. Additionally, this paper also extends the applicability to Electricity and Thermology. Overall, the task formation is very interesting and the idea is simple but effective.

**Summary Of The Review:**

As mentioned in my reviews regarding strengths and weaknesses, I think this paper has tackled a more practical and challenging problem than existing methods. The proposed method is also with novelty and soundness. Moreover, besides finding correct formulas in several known mechanisms, the authors further use it to discover new formulas for predicting real-world pedestrian dynamics. It seems that the obtained formulas are not only more precise (compared with a manually designed one) but also with good interpretability. Hence, I am leaning on the positive side.

---

> ### Author Response · Authors · 2022-11-19
> **Add a practical guide and random search experiment. (2/2)**
>
> About your suggestions on the clarity of this paper, we improve our paper based on your suggestion one by one as follows,
> **Q3.1**: Use a specific figure to demonstrate the design philosophy of the pruning-based search procedure.
>
> **Re**: Thank you for your valuable comment. We utilize Figure 11, 12 and 13 to show our design philosophy.
>
> Specifically, we first plot the learning curves of three cases w.r.t. different message-passing flows in Figure 11, corresponding to the one lacking the necessary message-passing connection, the ground-truth one, and the one with a redundant message-passing connection, respectively. We find that the ground-truth message-passing flow and the one with redundant message-passing connection have only little variations (the RMSE is around 0.1 and the MAPE is around 1.0) in performances after the loss function converges.
> However, the performance of message-passing flow lacking necessary connection decreases significantly (the RMSE is about 1.4 and the MAPE is 4.0).
>
> We further test the impact of the embedding size on the performance, where the learning curves are shown in Figure 12. When the embedding size is less than a certain number $2$, the performance decreases significantly (RMSE is more than 1.4), while the performance is similar (RMSE is around 0.2) otherwise.
>
> Last, we test whether the softmax function can learn the ground-truth aggregator. As we can see in Figure 13, the learning curve with softmax and with the ground-truth aggregator is quite similar, and we verify that the learned aggregator is the same as the ground truth.
>
> **Q3.2**: Give more details on the normalization process, as in “We normalize the input-output pairs to make the outputs have variance 1”.
>
> **Re**: In the training stage, we normalize the input-output pairs to make the outputs have standard deviation of $1$. Specifically, for each dimension of features, we divide the features by their standard deviation, obtaining the fitting errors with similar magnitudes. Since the complexities of different skeletons are also with similar magnitudes, the best value of $\lambda$ is similar for different datasets. This manipulation will not affect the constants in the formula, so when we retrain the constants, we will use the original input-output pairs. We add these descriptions in Appendix A.8 in the revised version.
>
> **Q3.3**: Make related work part shorter and save space for results analysis, especially the real-world example in A.4.
>
> **Re**: Thank you for your suggestion. Based on your comment, we shortened the related work part and moved the main content in A.4 into the main text (Section 3.2).
>
> **Q4**: A few typos. e.g., on page 8, “the time cost of each independent part is shown in Table 6” should be Table 3.
>
> **Re**: Thank you for the careful review. We have corrected your mentioned typos and thoroughly proofread the paper.

---

> ### Author Response · Authors · 2022-11-19
> **Add a practical guide and random search experiment. (1/2)**
>
> Thanks for your positive and insightful comments! We very much appreciate it. Please allow us to respond to your questions one by one as follows.
>
> **Q1**: For easy-to-use concerns, the paper should have a specific subsection that provides a practical guide on using the proposed method when discovering formulas in new physical domains. Currently, the related information is scattered in Sec. 3.1, Appendix A.3 and A.4, not concentrated enough.
>
> **Re**: Thank you for the helpful suggestion. Based on your suggestion, we add a subsection (Appendix B.2) to systematically introduce how to use our method and provide an algorithm (Appendix B.1) to show the whole process of running our model, making it easier for practitioners to use our model.
>
> When applied to a new physical domain, we can directly use the same DL-related hyper-parameters (such as the learning rate and embedding size of MLP as shown in Table 3) on old domains and slightly tune the search-related hyper-parameter (weight $\lambda$) near the previous optimal value on old domains (the stability of the proper $\lambda$ value is demonstrated in Table 5). Due to the randomness of DL training and the genetic algorithm, we train a model ten times with different seeds to get ten formulas and select the Pareto-optimal formula according to the score as the final formula. Overall, the above process in a new domain does not require heavy human labor.
>
> **Q2.1**: To better demonstrate the superiority of the designed pruning-based search procedure, authors can consider adding further studies in the appendix. For example, how about comparing with a random search method that does not decompose searching steps and does not leverage a pruning strategy?
>
>
> **Re**: Thank you for this helpful suggestion. Accordingly, to demonstrate the effectiveness of our searching method, we compare it with the random searching strategy and plot their searching processes in Figure 10 (Appendix A.7). We can observe that our method is much more efficient than the random search algorithm, suggesting that the searching problem is difficult and our method effectively reduce the original huge search space. Specifically, even if the random search algorithm takes 10 times longer than ours, the score of the best-searched skeleton is still 4.5 times worse than ours, and the searched skeleton is wrong.
>
> **Q2.2**: Also, Figure 6 can be improved by plotting four steps together, with the x-axis being absolute training time. This can make the explanation text much easier to follow.
>
> **Re**: Thank you for your helpful comment. Based on your comment, we plotted four steps together and added the absolute training time as the x-axis to better illustrate the search process as in Figure 6.

---

### Author Response · Authors · 2022-11-25
**General Comments**

We sincerely thank all the reviewers for their valuable comments.

Here, we would like to highlight the novelty, i.e., the difference between our work and SymDL, and insights from SR inspiring us to design our method.

***1. Difference on problem definition***

The difference between our work and SymDL is that SymDL additionally requires the skeleton of the formula, which can be automatically learned from data by our method. We further provide three examples to show their difference in three usage scenarios, which are shown in the figure in the following link,

https://anonymous.4open.science/r/GraphSR_rebuttal/README.md

(1) Left: both our model and SymDL can handle this mechanical scenario, while the skeleton is given in SymDL's setting and it is learned from data in our setting;

(2) Middle: our model can learn diverse skeletons from data in more different physical scenarios, such as circuit scenario, while SymDL require the user to know the knowledge about the formula skeleton;

(3) Right: Our model can learn the formulas for the scenarios where the formula skeleton is unknown, such as pedestrian dynamics, while SymDL cannot handle it.

***2. Difference on methodology***

Our key contribution is transforming the problem of learning formula skeleton to searching message-passing flow, a GNN architecture, so that we can utilize the idea of NAS for solving this problem of SR on graphs.

Technically, our search method design is inspired by the unique insight in this SR scenario, and is significantly different from existing NAS works. To demonstrate its effectiveness, we add a new experiment about the comparison with a random search strategy (Appendix A.7).

***3. Insights from SR***

The unique insight from SR inspiring our design of hierarchical pruning strategy from full GN is as follows:
The performance would be similar when the formula skeleton (message-passing flow) is a super-structure of the ground-truth one, while the performance degrades a lot if it is a sub-structure of the ground-truth one (shown in Figure 11 & 12).

---

### Decision · Program_Chairs · 2023-01-20

**Decision:**

Accept: poster

**Justification For Why Not Higher Score:**

This work is a decent generalization of Cranmer et al. (2020), but it does not fundamentally change how the community learns formulas.

**Justification For Why Not Lower Score:**

Learning symbolic formulas from physical systems is an important task and the authors did a good extension of Cranmer et al. (2020).

**Metareview: Summary, Strengths And Weaknesses:**

This work generalizes the  Cranmer et al. (2020)'s task of learning physical formulas with given an inductive bias of formula interactions in the form of a graph. The work proposes a method to learn graph-structured physical mechanisms from data using Pareto-optimal message-passing flows of GNN together with the symbolic models as components. Extensive experimentation shows the method to be significantly better than other state-of-the-art methods.

Reviewers generally agree that the work provides an innovative method to learn symbolic physics equations from data in interaction system induced by graphs. There were minor gripes with the comparisons with standard methods. But since standard methods do not generally have inductive biases on the interactions, they should fail (i.e., it not a fair comparison). If the authors nevertheless included them in the paper, it would significantly strengthen the argument (even though it is not a fair comparison).


**Note From Pc:**

if the above contains the word "oral" or "spotlight" please see: "oral" presentation means -> notable-top-5% and "spotlight" means -> notable-top-25%. As stated in our emails, we are disassociating presentation type from AC recommendations